# Combined ERT and GPR Data for Subsurface Characterization of Weathered Hilly Slope: A Case Study in Zhejiang Province, Southeast China

**Yajing Yan** [1] , **Yongshuai Yan** [2], **Guizhang Zhao** [3], **Yanfang Zhou** [4] and **Zhoufeng Wang** [5,*]

1    Collaborative Innovation Center for Efficient Utilization of Water Resources, North China University of Water Resources and Electric Power, Zhengzhou 450046, China; yajinghappy123@163.com
2    School of Resources and Environmental Engineering, Hefei University of Technology, Hefei 230002, China; 2020010056@mail.hfut.edu.cn
3    School of Geosciences and Engineering, North China University of Water Resources and Electric Power, Zhengzhou 450046, China; zhaoguizhang@ncwu.edu.cn
4    School of Foreign Studies, North China University of Water Resources and Electric Power, Zhengzhou 450046, China; zhouyanfang@ncwu.edu.cn
5    Key Laboratory of Subsurface Hydrology and Ecological Effect in Arid Region, Chang'an University, Xi'an 710054, China
*    Correspondence: wangzf@chd.edu.cn

**Abstract:** Rain-triggered landslides frequently threaten public safety, infrastructure, and the economy during typhoon seasons in Zhejiang Province. Landslides are complex structural systems, and the subsurface features play a significant role in their stability. Their early identification and the evaluation of potential danger in terms of the rupture surface and unstable body are essential for geohazard prevention and protection. However, the information about the subsurface acquired by conventional exploration approaches is generally limited to sparse data. This paper describes a joint application of ground-penetrating radar (GPR) with a 100 MHz antenna and the electrical resistivity tomography (ERT) method with the Wenner configuration to identify the stratum structure and delineate the potentially unstable body of a clay-rich slope, the results of which were further verified using borehole data and field observation. The acquired results from the GPR and ERT surveys, consistent with each other, indicate two stratigraphic layers comprising silty clay and silty mudstone. Moreover, the potential rupture zone very likely exists in the highly weathered mudstone in the depth range of 3–7 m, and the average depth is 5 m. In addition, the thickness of the unstable mass is greater on the east and crest parts of the slope. Conclusively, the optimum combination of ERT and GPR is reliable for conducting rapid and effective delineation of subsurface characteristics of clayey slopes for risk assessment and mitigation during the typhoon season.

**Keywords:** hilly slope; electrical resistivity tomography (ERT); ground-penetrating radar (GPR); subsurface structure; potential sliding surface; Zhejiang; typhoon





## 1. Introduction

Zhejiang Province, an economically developed and densely populated region in the subtropical zone of China, is exposed to a high risk of rainfall-triggered landslides caused by an incremental occurrence of extreme weather events [1–5]. Moreover, the province's landscape is dominated by mountains and hills, accounting for about 75% of its total area, which can easily lead to landslides. Additionally, increasing demand for modern infrastructure has caused more engineering disturbance, thus multiplying human risks. At present, the occurrence of landslides is still one of the greatest threats to local inhabitants and infrastructure, as exemplified by the Xiashan village landslide in 2001 [4], the Lidong village rockslide in 2015 [6], the Sucun village rockslide in 2016 [7–9], and the Shanzao landslide in 2019 [10]. Currently, many slopes are still slide-prone, the majority of which

are typically small in volume. It is a time- and resource-consuming task to thoroughly investigate them. According to Mccann and Foster [11], estimation of landslide stability has to consider the definition of the 3D shape of the unstable body with particular reference to the failure surface. Hence, there is a pressing need for developing and implementing actions for the accurate and rapid identification of subsurface features of natural slopes in Zhejiang.

As regards the practical techniques of identifying subsurface objects, conventional geotechnical (e.g., drilling, tunneling, and trenching) and geophysical approaches (e.g., electrical, electromagnetic, and seismic methods) are the most known ones, which have been broadly applied around the world. The former geotechnical approaches allow a detailed subsurface description at sparse locations, but they fail to delineate continuous spatial information. Moreover, they are very costly and relatively time-consuming. Indeed, underground materials often show high lithological and tectonic variability within short distances. However, the geophysical techniques, which are flexible, relatively quick, and deployable on slopes, can provide bulk spatial data directly or indirectly linked with the lithological, hydrological, and geotechnical characteristics of unstable slopes [11,12], therefore providing continuous geophysical mapping of subsurface features, and overcoming the drawbacks of geotechnical measurements.

Among all geophysical methods, the electrical resistivity tomography (ERT) and ground-penetrating radar (GPR) methods have proven to be highly efficient approaches in landslide research [13]. By analyzing the reflected signal of transmitted waves from the interface where there is a difference in materials, the GPR technique can image shallow subsurface structures (even small cavities) and determine the distance at which they are located. Since the 1980s, GPR has been increasingly accepted for the localization of fractures or cracks [14–18], the identification of stratigraphy or shear deformation [19–23], and the characterization of soil water variations [24–27] in the geological, environmental, and engineering areas. However, the transmitted waves of GPR are strongly attenuated in conductive zones (e.g., water-rich, clay-rich) and cannot penetrate at greater depths to identify unknown objects of interest. For a slope with high vegetation coverage, this phenomenon might be further exacerbated. With a lower resolution and greater penetration depths than GPR in conductive environments, the ERT technique has been used widely for various landslides, from rockslides to debris slides, in different geological environments from rock to soil materials, to identify the slip surface and hydrological conditions, depict the internal structures, monitor the movement, and disclose the underground faults and cracks [28–34]. Falae et al. [31] discussed the recent trend in applying ERT in landslide studies. This method relies on measuring the electrical properties between two electrodes when transmitting a pulsed current between two other electrodes, which allows for characterizing the unsteady body compared with the material having different electrical potentials.

Given the non-uniqueness of dataset interpretation, and the drawbacks of individual techniques in resolution and penetration depth, GPR and ERT have been jointly used for the investigation of subsurface features thanks to their complementarity [35–43]. Jongmans and Garambois [28] concluded that almost all the advantages of the geophysical method corresponded well to the disadvantages of the conventional geotechnical techniques. Perrone et al. [13] stated that the joint application of GPR and ERT could solve and overcome the resolution problems of every single method. Specifically speaking, GPR provides more helpful information on the shallow layers, while ERT is preferable for the intermediate–thick layers. The combination of the above-mentioned methods is therefore believed to have the potential to become a valuable tool for the pre-evaluation of high-risk sliding areas. However, more attempts seem to be necessary regarding its accuracy and applicability when probing clayey slopes.

For these reasons, this work aimed to test the ability of the joint use of ERT and GPR to distinguish the subsurface characteristics of a clayey slope, and to discuss the optimum combination. Two geophysical measurements were performed along with three profiles, at three sites where three boreholes are also available. Validated and calibrated with borehole data and field observation, the unstable body and potential slip surface could be sufficiently inferred. The effectiveness and limitations of ERT–GPR surveys for fast characterization of the subsurface are also highlighted. The novel aspects of this study are as follows: (1) it was conducted on a clay-rich slope which is usually not friendly for GPR surveys to distinguish different stratum layers (clay and mudstone); (2) variations in amplitude and energy with depth for three single-channel GPR waves are also analyzed, unlike in previous similar studies focusing on the GPR profile; (3) it optimizes the typical GPR profile superimposed with the elevation level; (4) a 3D model of the potentially unstable body is drawn for direct visualization. These findings provide a reliable alternative for a more comprehensive and faster investigation of active slopes in regions where typhoons are frequent and unstable terrain is abundant.

## 2. Materials and Methods

### 2.1. Site Description

The studied slope (27°34′59″ N, 119°54′10″ E) is located in Yuxi village in Taishun County, Wenzhou (Figure 1). It is about 300 km away from the capital city of Hangzhou, in the southeast of Zhejiang Province, which belongs to a subtropical marine monsoonal region with average annual precipitation of about 2000 mm. Moreover, the distribution of rainfall over a whole year is not uniform. About 71.2 percent of the rainfall events concentrate from May to September due to the influence of monsoons or typhoons, during which geological disasters are highly likely to occur. For example, the super typhoon Lekima in 2019 induced more than 400 landslides, debris flows, and numerous unstable points, including the famous Shanzao rockslide causing 32 casualties in Wenzhou city, according to the government report of Zhejiang Province [10]. In the future, climate change will continue to exacerbate the frequency and intensity of disasters in China [44] (p. 14).

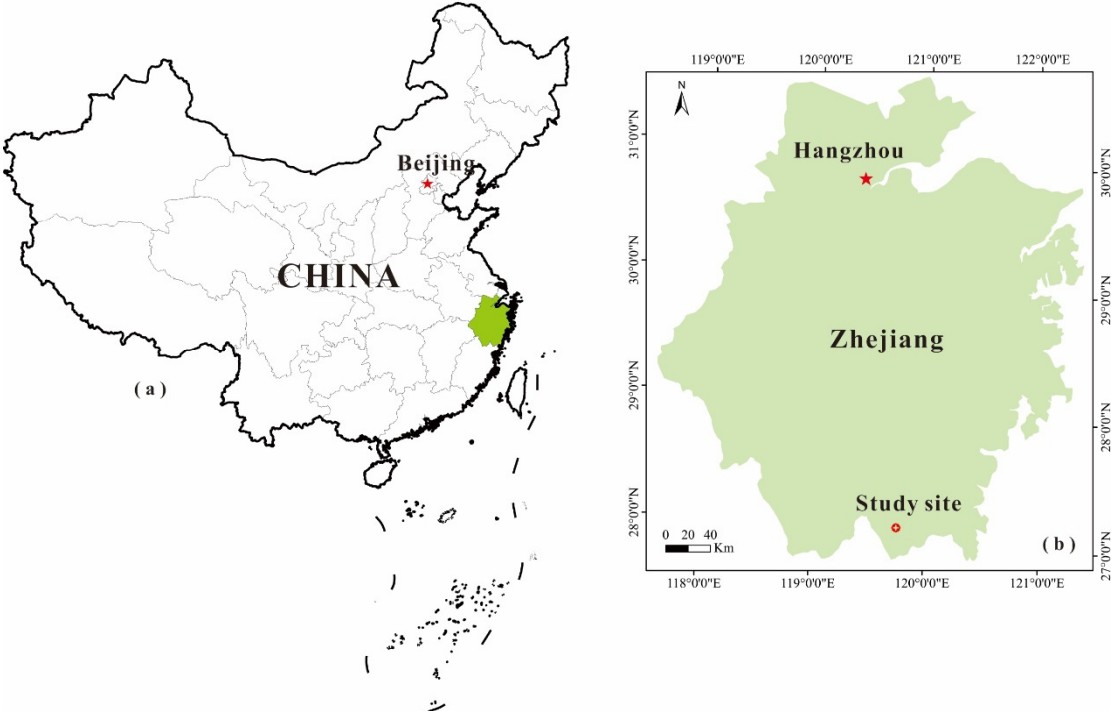

**Figure 1.** Location of the study site (red circle) in Zhejiang Province. (**a**) Map of China and its capital city Beijing, and of (**b**) Zhejiang Province and its capital city Hangzhou.

In Taishun County, more than 200 landslides have occurred since 2001 [45], in which small and shallow movements were common, putting a great strain on the county's people, resources, and environment. Although no fatal landslide disasters have been recorded, all villages in this county are still, to varying extents, facing negative impacts from geological disasters as a result of climate change, rapid economic growth, and urbanization.

Geomorphologically, the entire terrain of the slope is inclined from the south toward the north and drops in a step form due to artificial agricultural activities for the bayberry plant, having an average slope gradient of 30° and an original dip of 5° in the NE direction (Figure 2a). The area of the study site is a hilly terrain, the elevation of which varies between 370 and 402 m a.s.l. Geologically, its stratum units were precisely disclosed using rock and soil samples collected from boreholes, whose locations and details are shown in Figure 3, and further verified by stratum outcropping (Figure 2c). The lithological properties of different layers were disclosed by drilling core samples, as shown in Figure 4. There are two primary stratum layers in the sedimentary succession according to the borehole information. At the base of the stratigraphic column, silty mudstone of the lower Cretaceous Guantou Group ($K_1g$) is overlain by a 1–5 m-thin Quaternary (Q) soil, containing strongly weathered and fresh rock. More precisely, the near-surface layer is mainly composed of silty clay with granular gravels, the majority of which are loose colluvial sediments with large pores and high porosity. Over 80% of the landslides in Zhejiang Province occur along the colluvium–bedrock contact resulting from the varying soil moisture and pore water pressure of the colluvial deposits [46]. The upper grayish $K_1g$ mudstone is strongly weathered and fractured, while the bottom is rarely weathered or fresh. The slope's potential failure will probably develop between these two stratum layers.

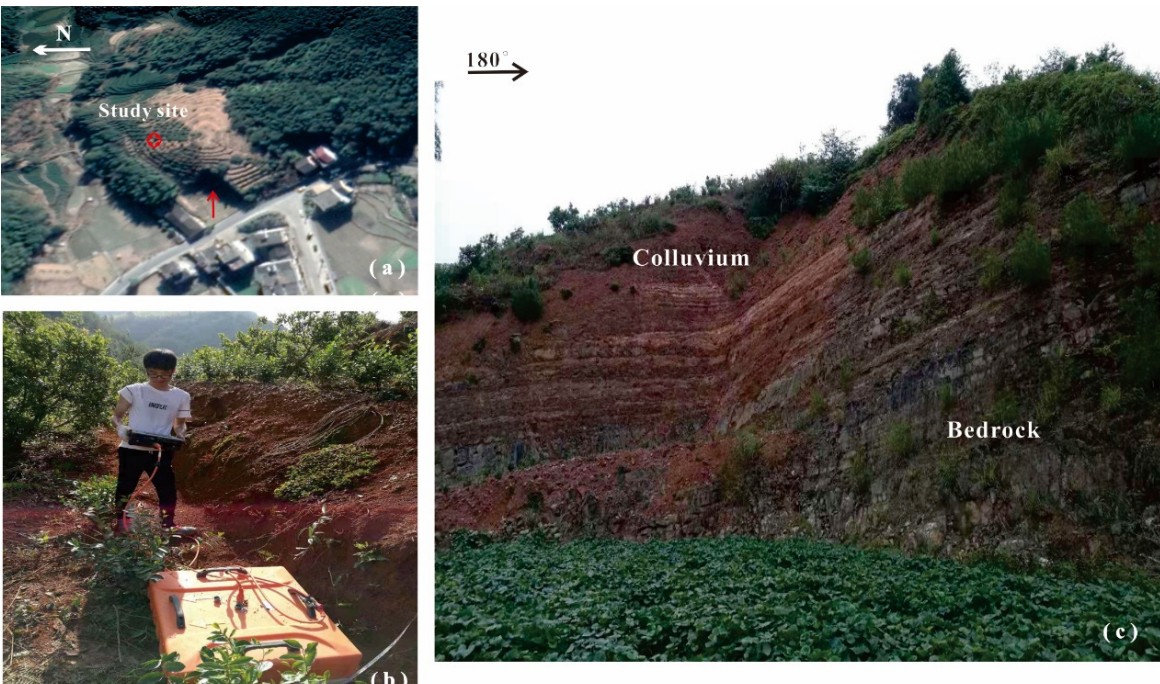

**Figure 2.** (**a**) Photo of the study site from the air. The red circle marks the study site, and the red arrow represents the direction of the camera that obtained the photo shown in (**c**). (**b**) Photo of the acquisition work showing the GPR system, step-shape landform, and vegetation coverage. (**c**) Field observation of the stratum.

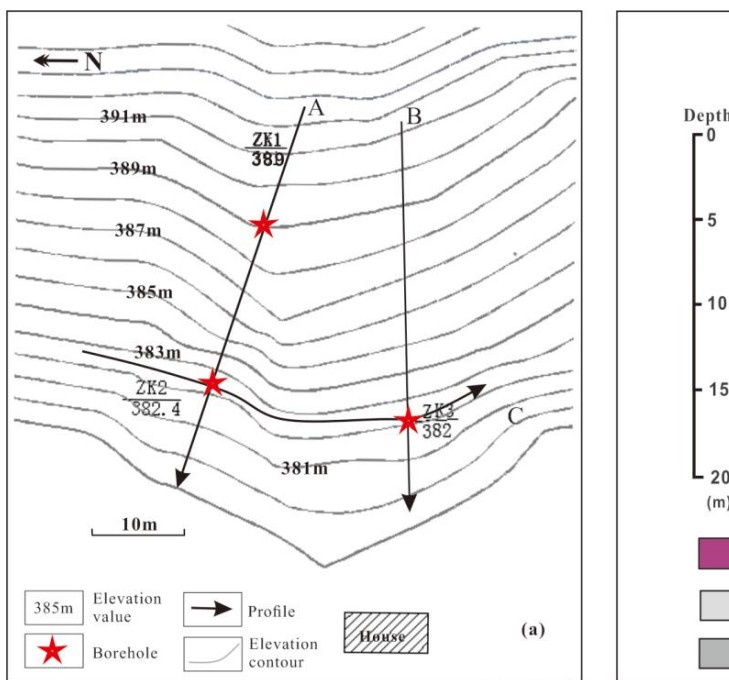
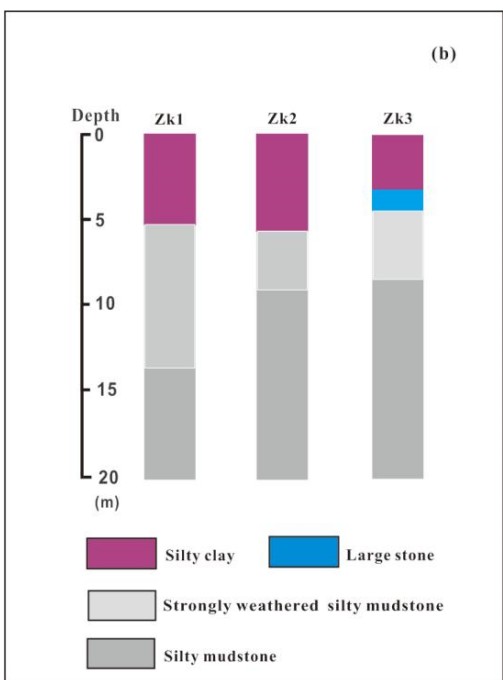

**Figure 3.** (**a**) Locations of geophysical survey lines (A, B, C) and boreholes (ZK1, ZK2, ZK3). (**b**) Drilling core logs of the three boreholes (ZK1, ZK2, ZK3).

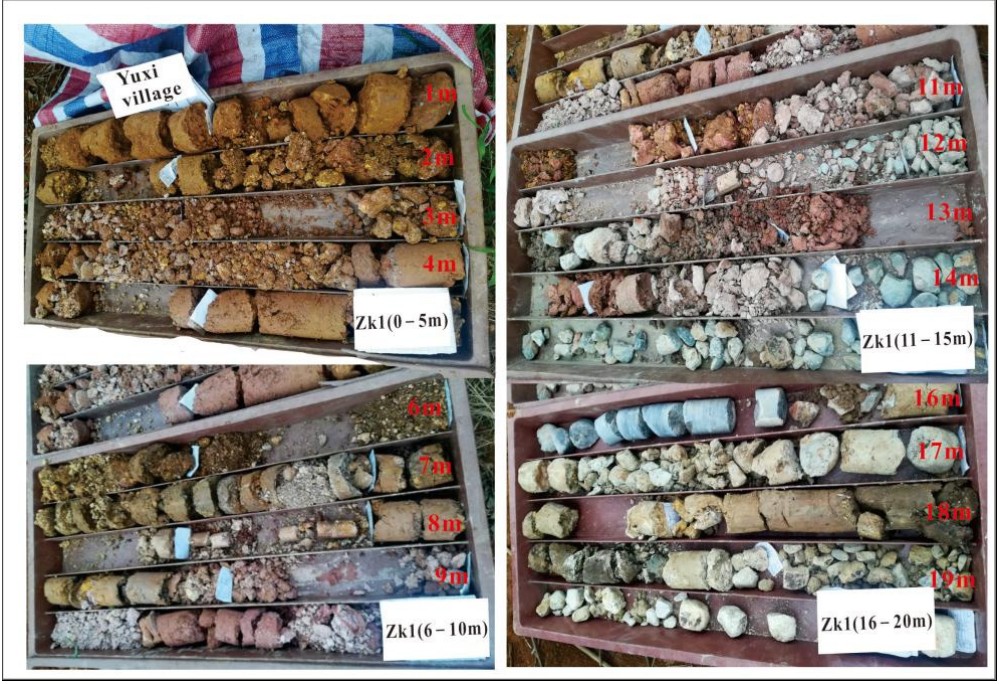

**Figure 4.** *Cont.*

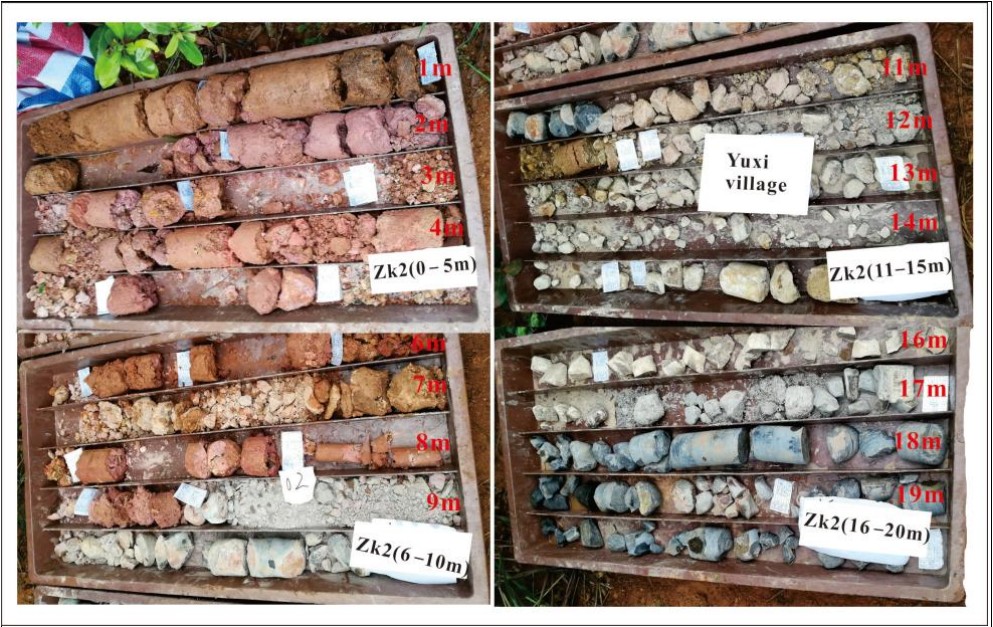

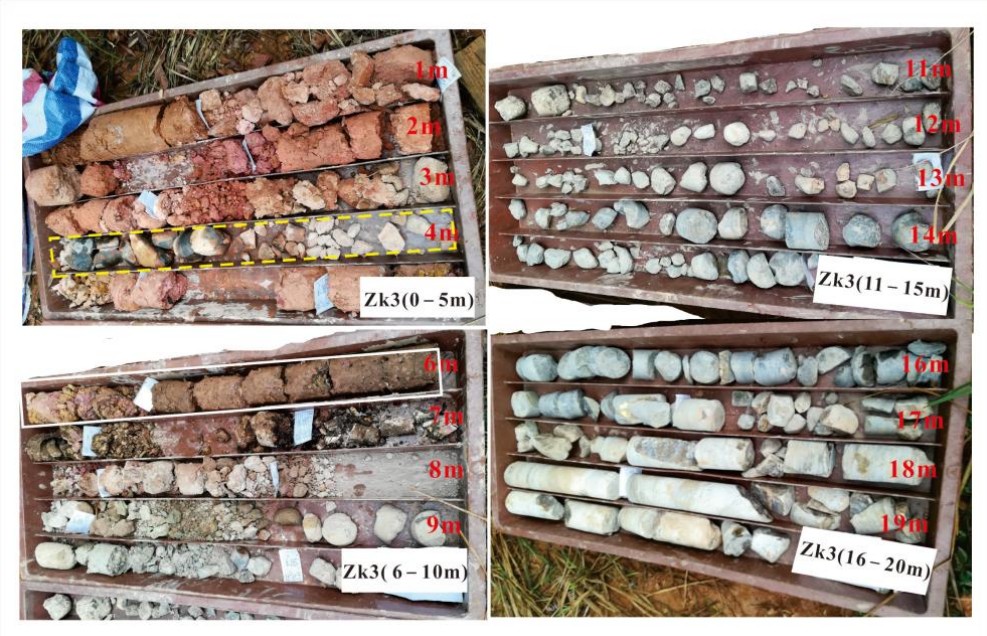

**Figure 4.** Drilling core samples from three boreholes (ZK1, ZK2, ZK3).

During drilling campaigns, soil and rock samples from three boreholes were carefully collected and transported to the laboratory for basic physical properties, mineralogical compositions, and direct resistivity measurements. The Atterberg limits and particle size distribution of the soil samples were determined following the Chinese National Standards (CNS) GB/T50123-2019 [47] (pp. 29–43). The bulk mineralogy and clay mineralogy of the rock samples were measured using an X-ray diffractometer following the methods described by Moore and Reynolds [48] (p. 378). More specifically, clays in the soil samples ranged from 41 to 47%, silts from 27 to 30%, sands from 20 to 25%, and gravels from 4 to 7%. Their liquid limits and plastic limits were in the range of 39.7 to 42.4 and 20.8 to 23.2, respectively (Table 1). The results of the semi-quantitative XRD analysis show that clay minerals (50–52%) and quartz (34–36%) were the primary minerals, followed by mica and feldspar, in all rock samples. In terms of the clay mineralogy, these samples were primarily

composed of illite, with some chlorite and kaolinite (Table 2). Apparently, the slope under probing is a clay-rich body.

**Table 1.** Basic physical properties of the soil samples.

| Soil Sample | Atterberg Limits (%) | | Particle Size Distribution (mm, %) | | | |
| | Liquid Limit (%) | Plastic Limit (%) | Clay (<0.005) | Silt (0.005–0.075) | Sand (0.075–2) | Gravel (>2) |
| --- | --- | --- | --- | --- | --- | --- |
| ZK1 | 42.4 | 23.2 | 47 | 28 | 20 | 5 |
| ZK2 | 39.7 | 20.8 | 41 | 30 | 22 | 7 |
| ZK3 | 40.9 | 21.5 | 44 | 27 | 25 | 4 |

**Table 2.** Mineralogical compositions and resistivity of the rock samples.

| Rock Sample | Whole Rock (%) | | | | Clay (%) | | | Resistivity (Ωm) | |
| | Quartz | Feldspar | Mica | Clay | I | K | C | Weathered | Fresh |
| --- | --- | --- | --- | --- | --- | --- | --- | --- | --- |
| ZK1 | 34 | 8 | 6 | 52 | 56 | 23 | 21 | 785 | 1432 |
| ZK2 | 36 | 8 | 5 | 51 | 55 | 26 | 19 | 801 | 1365 |
| ZK3 | 34 | 7 | 9 | 50 | 53 | 22 | 25 | 812 | 1309 |

Note: I—illite; K—kaolinite; C—chlorite.

The studied slope threatens the safety of county road No. 601 and the safety of 15 residents in 5 households, with a potential economic loss exceeding CNY 1 million. Furthermore, it was chosen because of its clayey conditions and typical geological structure (two stratum layers: unconsolidated topsoil and bedrock), representing the most common type of landslide that has not received sufficient attention in Zhejiang Province. There are the upper loose topsoil layer allowing infiltration of rainwater, and the relatively dense bedrock layer serving as an aquiclude. Usually, the contact zone between these two layers has a high likelihood of developing into the rupture surface or area in intensive rainstorms.

*2.2. Methodology*

In collaboration with drilling core samples, we combined ERT and GPR to identify the subsurface features of the studied slope, in particular with delineation of the shape of the potential unstable body and potential slip surface, which were tentatively assumed to be fault and/or joint planes, and interfaces between topsoils and/or highly fractured rocks with intact bedrock [49]. All geophysical tests were conducted following the Standard of Ministry of Water Resources of the People's Republic of China: SL/T 291.1-2021 [50] (pp. 6–10, 15–56, 143). Information about ERT profiles A, B, and C is shown in Figure 2. In what follows, the details of the ERT and GPR measurements are described.

2.2.1. Data Acquisitions

Based on Ohm's law, ERT measurements are accomplished by measuring electrical potentials (ΔV) between an electrode pair while introducing a direct current (I) between another pair of electrodes [51] (p. 806). Then, the resistive properties of the underground medium can be acquired, commonly expressed in the form of apparent resistivity. In this study, acquisition was carried out using the DUK-2B high-density resistivity imaging system with 64 electrodes spaced by 1 m from Chongqing geological instruments. Two-dimensional (2D) resistivity profiles along three survey lines (labeled A, B, and C) were obtained using the Wenner configuration. The length of the survey lines varied from 51 m to 60 m, while the expected investigation depth was between 11 m and 20 m, correlating with the elevation and topography conditions of the studied slope. Profiles A and B (longitudinal profiles), striking SE and SW, were obtained down-slope, while profile C (cross-profile) was obtained perpendicular to the dip direction of the slope.

Three drill holes (ZK1, ZK2, and ZK3) were created using a mobile drilling rig installed on a truck for stratum identification, which could provide information about the lithology up to the depth of 25 m based on the extracted core samples. The exact locations of these boreholes are shown in Figure 3.

Founded on Maxwell's equations, GPR measurement can be conducted by transmitting pulses of high-frequency electromagnetic (EM) waves that travel through the subsurface, and receiving EM signals by receiver antennas. When EM energy emits into the ground, these EM waves can encounter different interfaces, resulting in part of the GPR signals being reflected to the receiver [52]. These interfaces commonly represent changes in the physical properties of the subsurface system that are expressed by contrasts in relative dielectric permittivity. Subsequently, waveform depth, time–frequency, and amplitude characteristics of the reflected signals generated at the interfaces are collected and then analyzed. In the pioneer studies (summarized in [28]), it was concluded that an antenna with a higher frequency corresponded to a higher resolution and smaller penetrating depth. A 100 MHz antenna commonly has a higher resolution to image small features up to the depth of 4–5 m and a lower resolution up to a depth of 20 m. In order to ensure a sufficient detecting depth and resolution at the same time, the ground-penetrating radar SIR-3000 from Geophysical Survey Systems, Inc. (GSSI) Co., Ltd. a was employed at three measured points close to the three boreholes, along with three profiles similar to the ERT lines, coupled to a 100 MHz antenna with an average penetrating depth of 15 m. Based on GPR datasets, the depth of unknown targets transformed from the travel time was calculated using a GPR wave velocity of 0.12 m/ns, estimated using the borehole data. The calculation process was demonstrated by [36]; no more details are presented herein.

### 2.2.2. Data Processing

The collected resistivity datasets along all profiles were inverted using the programming code and algorithms of the software RES2DINV, developed by [53]. The 2D resistivity images were obtained by performing elimination of bad data points, topographic correction, an RMS convergence restraint, least squares inversion, and a robust smoothness constraint [53]. All of the resistivity profiles were created after a maximum of 8 iterations, and the RMS error in percentage of the last iteration was controlled to lower than 3%, indicating the good and reliable results of the ERT surveys.

Radar data processing was accomplished using RADAN (version 6.0, Geophysical Survey Systems Inc., Nashua, NH, USA), including static corrections, background removal, distance norm, range gain, finite impulse response (FIR) filtering, deconvolution, and time–depth conversion.

## 3. Results and Discussion

In what follows, information about the subsurface in terms of the amplitude/energy of a single-channel radar wave, inverted resistivity section, and GPR reflection profile is presented in Figures 5–9 and further verified using borehole data and field observation.

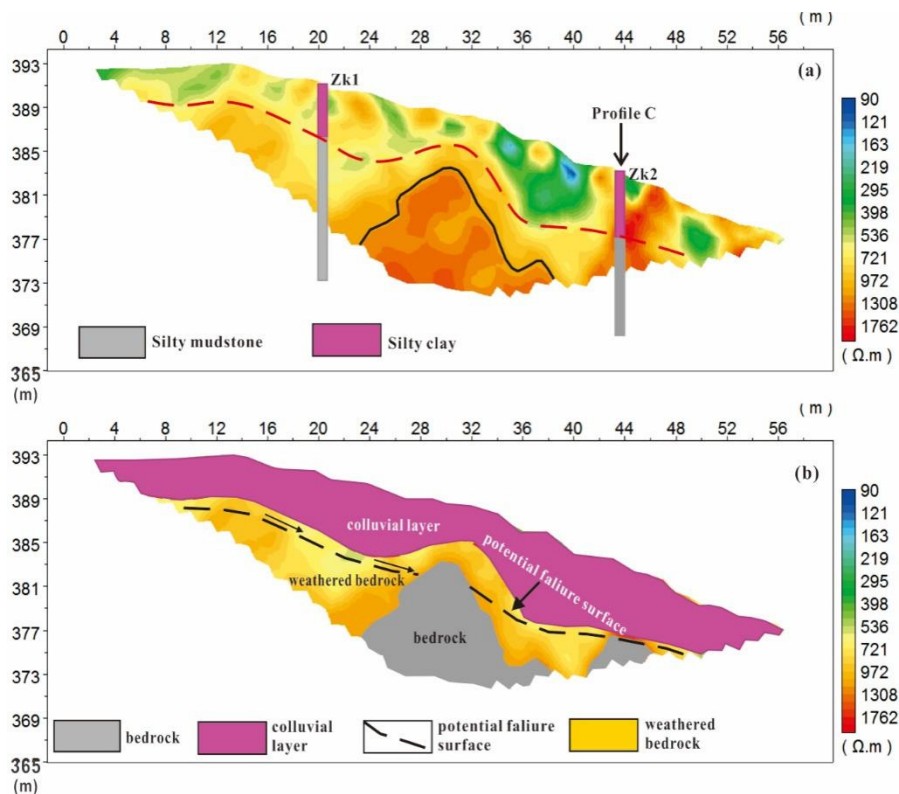

**Figure 5.** ERT profile A and its geological interpretations. (**a**) ERT image of profile A, (**b**) geological interpretations from ERT profile A.

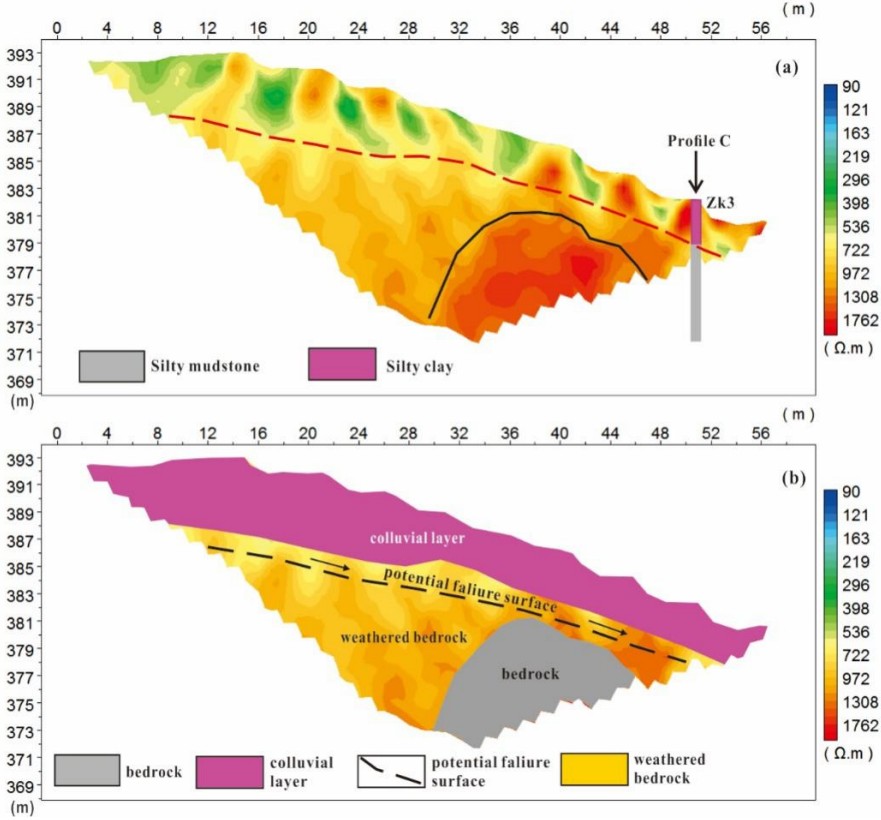

**Figure 6.** ERT profile B and its geological interpretations. (**a**) ERT image of profile B, (**b**) geological interpretations from ERT profile B.

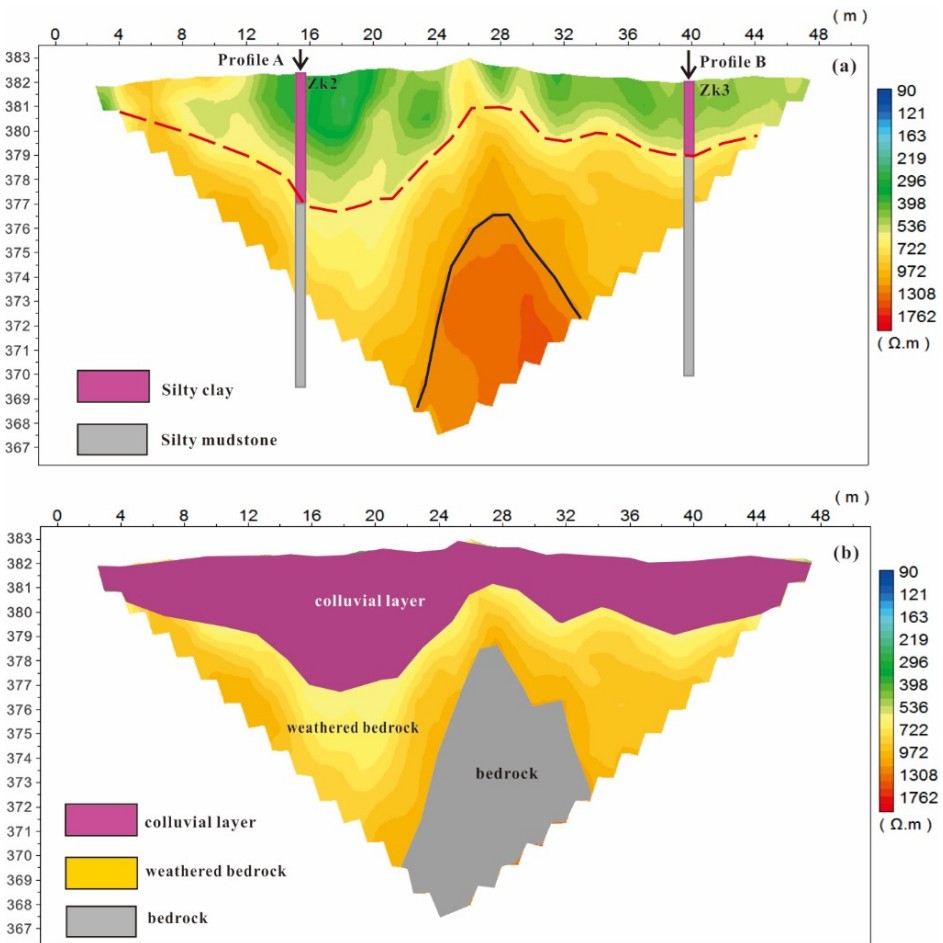

**Figure 7.** ERT profile C and its geological interpretations. (**a**) ERT image of profile C, (**b**) geological interpretations from ERT profile C.

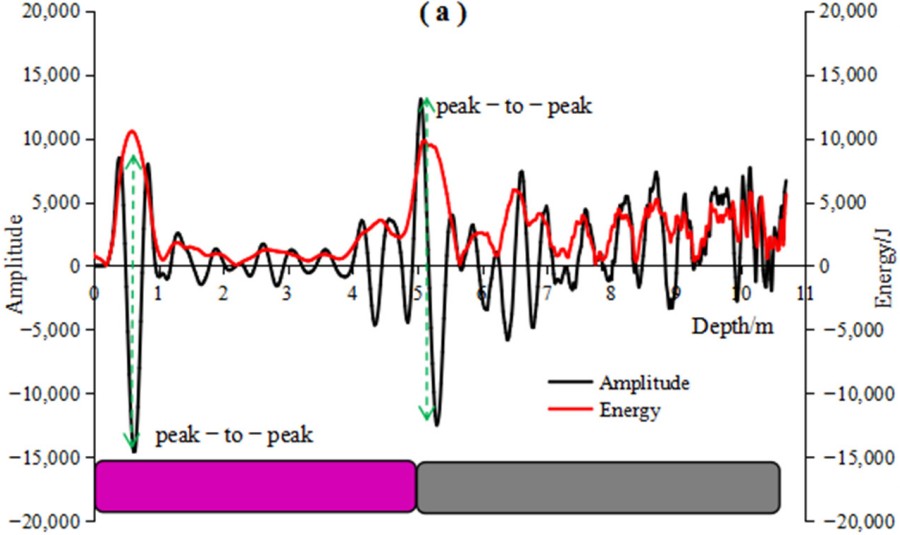

**Figure 8.** *Cont.*

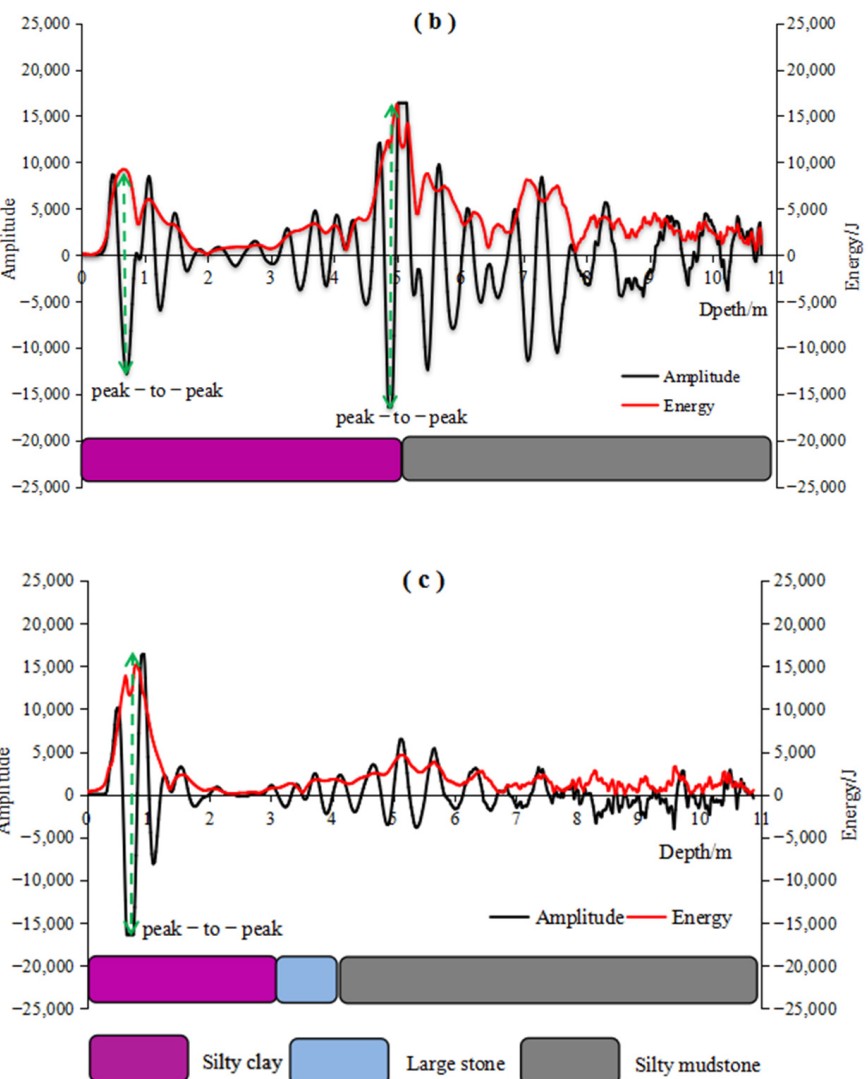

**Figure 8.** Waveforms of a single-channel GPR wave at three sites near the three boreholes, showing variations in amplitude and energy with depth. The black solid lines represent the amplitude, while the red solid lines denote the energy. The green dashed line with two arrows corresponds to the peak-to-peak phenomenon. (**a**) Waveform of ZK1, of (**b**) ZK2, and of (**c**) ZK3.

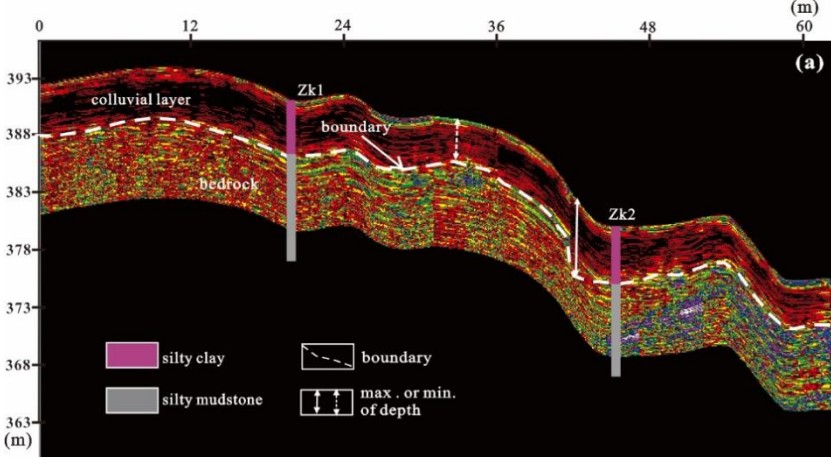

**Figure 9.** *Cont.*

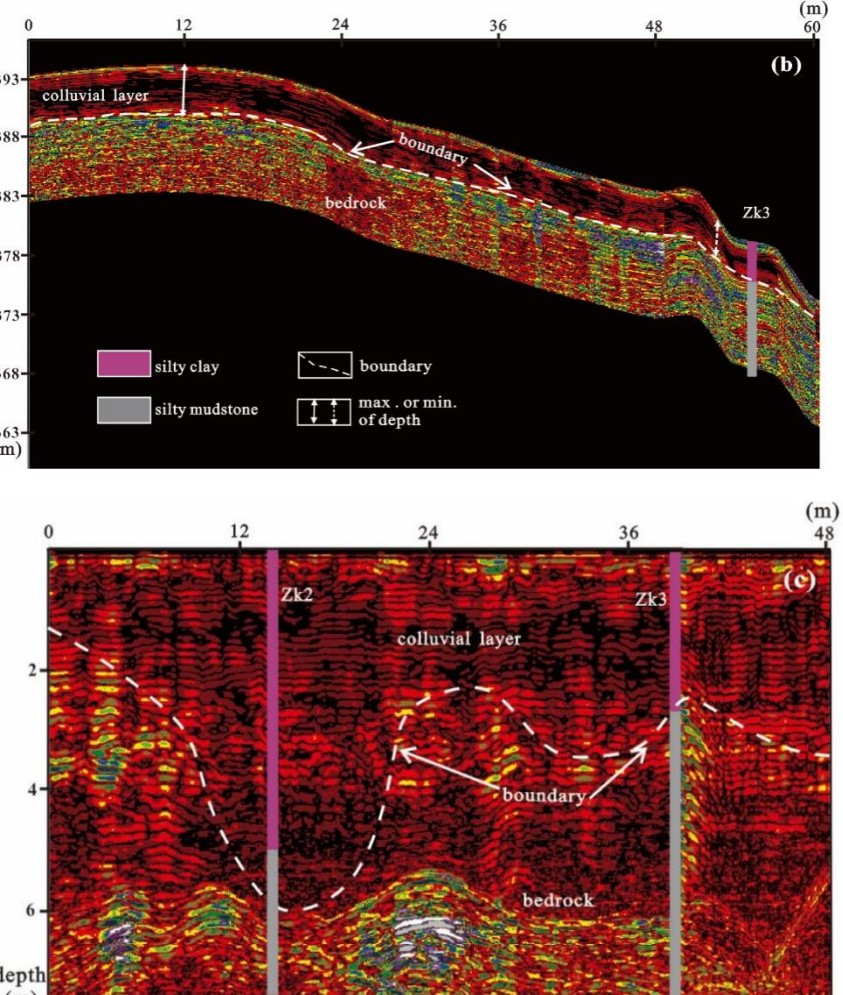

**Figure 9.** GPR image profiles. (**a**) Profile A superimposed elevation values (*Y* axis), (**b**) profile B superimposed elevation values (*Y* axis), and (**c**) profile C at the elevation of approximately 384 m. The white dashed line indicates the boundary between different geological layers, while the white solid and dashed line with two-way arrows in (**a**,**b**) represents the maximum and minimum thicknesses of the overlying layer.

### 3.1. ERT Profiles and Interpretations

To obtain electrical properties of the subsurface, ERT surveys along three profiles (A, B, and C) were carried out to image inverted resistivity variations with depth in autumn at the end of the wet season. The inverted resistivity sections and their corresponding geological interpretations are described in Figures 5–7, respectively. Of particular note is that profiles A and B superimpose the elevation values, while profile C is a raw image. Boreholes ZK2 and ZK3 were located at the bottom of the slope, while ZK1 was laid near its crest. The effective probing depth of ERT was in the range between 12 m and 20 m, which was greatly dependent upon the electrode spacing, resistivity contrast, broken properties of the medium, and landform of the slope [54]. Due to its step-shape landform, the variations in resistivity might also be impacted by the installed location of the electrodes, besides the mineral components, underground water, and its chemical composition and temperature.

The resistivity range measured in this study was between 90 and 1762 Ohm.m, and its distribution pattern can provide information about the stratigraphic distribution. The ERT images from Figures 5–7 with the constraints of geological field observations (Figure 2c) and drilling core samples (Figure 4) highlight two main resistive zones and their corresponding stratigraphic layers in all probing profiles. In profile A (Figure 5), the near-surface layer,

with a variable thickness between 1 and 7 m, has a relatively low resistivity distribution (<722 Ohm.m), which can be associated with the overlying colluvium composed of silty clay with granular gravels. The underlying layer at a depth ranging between 8 and 20 m is primarily characterized by relatively high resistivity values (>722 Ohm.m) and can be related to the bedrock consisting of silty mudstone. Meanwhile, the average thickness of the colluvial layer is about 4 m. The colluvium thickness approaches its maximum at the location of 38 m, but it goes down to its minimum on the middle and crest parts of the slope, indicating the fluctuation in the bedrock surface. Additionally, a higher-resistivity body (>1308 Ohm.m) exists in the range between 22 and 38 m at an elevation level of about 383 m, which most likely represents the rarely weathered bedrock by comparing with ZK3 that is the closest exposure site of the subsurface medium. Differently, the thickness of the colluvial layer in profile B decreases gradually from the crest to the toe of the slope, with an average value of 3 m, ranging from 2 m to 6 m. According to Figure 6, the depth to the bedrock is the smallest at the position between 48 and 58 m. Similarly, there is also a higher-resistivity zone delimited by the black thick line existing in the right part of profile B at an elevation below 382 m. These findings indicate that a rugged bedrock surface does not exist in profile B in the range of the horizontal distance.

Since profile C has approximately an EW routing direction at an average elevation level of about 384 m, it is roughly perpendicular to the slope's other two ERT profiles. Its inverted section evidences a fluctuating cover of colluvial materials, which further supports the interpretations of the other ERT profiles. It is clearly seen in Figure 7 that the colluvium thickness of profile C is about 5 m near the position 15.3 m, decreases to <1 m at the point of 28 m, and then increases to 2.5 m at its right edge, in good agreement with those variations in profiles A and B. Apparently, the colluvial material cover is non-uniform on this slope, implying the limitations of drilling or trench technology at several points.

In short, the bedrock surface is rugged with obvious undulation on the east part of the slope, while it follows the topography without any noticeable fluctuation on its west part. The transition zone or interface between the silty clay and weathered mudstone layer could be interpreted as the potential failure zone or surface. Similar interpretations were also made by many researchers based on ERT results [32,36,41]. In the literature, ERT has been used in geology, where [36] summarized the resistivity values of common materials, with the resistivity value being about 10–2000 Ohm.m for silts, 1–100 Ohm.m for clay, 100–1400 Ohm.m for gravel, and >1000 Ohm.m for rocks. Similar to the values reported by [54,55], herein, the resistivity of <722 Ohm.m for the overlying layer composed of gravelly silty clay acquired from ERT matches well with a mixture of silt, gravel, and clay materials. Consistent with the direct value from the laboratory tests (Table 2), the average resistivity of bedrock from ERT was about 1000–1100 Ohm.m, and it reduced to the range of 722–1000 Ohm.m in the upper weathered zone, probably due to the increased content of finer and clayey particles resulting from the weathering of the mudstone. The same variations in resistivity between weathered and fresh rocks are observable in previous studies [32,54].

To further validate the accuracy and effectiveness of the ERT results, GPR measurements involving three lines that are the same as the ERT arrangements and three other points near these three boreholes were carried out. Therefore, in what follows, the GPR results are systematically provided and discussed.

### 3.2. GPR Results and Interpretations

The GPR results of three surveying lines (A, B, and C) and three probing sites close to the boreholes as well as their main interpreted features are presented in Figures 8a–c and 9a–c, respectively.

### 3.2.1. Waveforms in Depth Domain and Interpretations

To directly depict the propagation process of EM waves, GPR signals containing amplitude and energy information are exhibited firstly in the depth domain based on an

estimated velocity of 0.12 m/ns, which was determined by measuring the travel time in a known borehole. In Figure 8, the variations in the amplitude and enemy of the EM waves with depth are revealed by analyzing three single-channel EM waves acquired from three probing sites close to Zk1, Zk2, and Zk3. It should be pointed out that errors in the depths obtained using GPR and boreholes are inevitable due to changes in the water content in subsurface materials.

In all profiles, the waveform in the depth domain highly assists in delineating the lithological boundaries up to a maximum depth of 12 m in Figure 8. When the peak amplitude of the radar wave represents an interface between layers or mediums with different dielectric properties during propagation [55], as can be seen in Figure 8a,b, the amplitude variations (black line) of the EM wave exhibit two significant peak-to-peak phenomena around the depths of 0.5 m and 5.1 m for ZK1, and 0.5 m and 4.9 m for ZK2. Let us compare the waveform with the geological stratigraphy (colorful legends). The first peak-to-peak amplitude is highly related to the air–ground interface at a depth of about 0.5 m, and the second peak-to-peak position can be interpreted to represent the topsoil–bedrock interface at a depth of 4.9 m for ZK1 and 5.05 m for ZK2. As the energy of the EM wave is proportional to the square of the amplitude, it exhibits consistent undulating trends similar to the amplitude with depth, having two principal peak values at similar positions (red line). It is thus believed that the amplitude and energy of radar waves could be in a relatively stable state when propagating in the same medium, whereas they fluctuate dramatically around these interfaces.

However, for the GPR signal of ZK3 in Figure 8c, the amplitude displays only one peak-to-peak phenomenon at a depth of about 0.5 m, and it thereafter shows a decreasing trend with slight fluctuation. Correspondingly, the energy reaches a peak value around the 0.5 m depth, and afterward, it shows little change with the increase in depth. For this surveying site, the interpretation of the amplitude and energy dataset does not coincide well with the borehole data. This contradiction may be mainly because of the fact that broken anomalous rock bodies exist in the depth range between 3 and 4 m. Generally, the fragments of rock existing in the underground medium are more likely to produce interference signals, during which the GPR signals are most attenuated. The observations made in drilling hole ZK3 delimited by the yellow dashed line in Figure 4 further confirm the presence of an altered rock (fractured) zone.

Apparently, the amplitude/energy–depth curves obtained from sites near ZK1 and ZK2 distinctly show two similar relatively stable stages separated by a transition zone, implying two layers with different dielectric constants of mediums at the probing points. However, the anomalous bodies (large stones or boulders) in geological units possibly contribute to the strong attenuation of electromagnetic waves [41], leading to the reduction in the energy and amplitude of the radar signals. This explains why the amplitude and energy of the point close to ZK3 gradually decreased with a slight change. In other words, the broken properties of the subsurface medium except for the dielectric capability also contribute to the stronger reflection, refraction, and diffraction behaviors of the radar waves. This indicates that the variation in the amplitude or energy of the radar waves with depth is also suitable for identifying the internal structure of slopes, whereas its applicability may be worse in more broken masses.

### 3.2.2. GPR Reflection Profiles and Interpretations

Generally, GPR signals of geological formations vary mainly depending on the nature of the components and their textures, and banded sediments often present superimposition of layered reflections [36,41]. Overall, the radargram in Figure 9 allows the characterization of heterogeneous deposits in the slope, and it contains abundant GPR signals marked with different colors representing various amplitude values. In this study, the richer the color, the richer the reflection, and vice versa. Accordingly, all GPR images indicate a similar reflected pattern over the topsoil versus the bedrock.

Figure 9a explores subsurface information up to 13 m depth, and two distinctive reflected zones are observed in profile A. The area along the upper part above the white dashed line showing darker and continuous to moderately continuous reflectivity corresponds to the colluvial layer composed of gravelly silty clay, as disclosed by ZK1 and ZK2. However, it is essential to note that the top area of the colluvium ranging from the surface to about 1 m depth presents brighter and discontinuous reflectors. The effect of the complex root systems of plants could explain this colorful phenomenon. Below this layer, the abundant (colorful) and discontinuous reflectors most likely illustrate the strongly reflective behaviors of the GPR waves, which are interpreted as highly fractured or strongly weathered silty mudstone bedrock. The maximum and minimum depth values to the bedrock surface occur at 33 m and 42 m, respectively, agreeing with the occurrences in ERT profile A (Figure 5a). Additionally, the lithological samples collected from the drilling holes provide further support for the interpretations of GPR profile A.

The GPR investigation of profile B (Figure 9b) reached a maximum depth of 11.5 m, presenting two primary radar reflected zones. The overlying area is expressed by moderately continuous darker reflectors with some occurrences of brighter discontinuous reflectors in the depth range of 0–1 m, having a thickness range between 1 and 5 m. This is highly likely related to the gravelly silty clay layer. A number of brighter discontinuous reflectors are detected within the underlying layer, which corresponds to the bedrock of the strongly weathered silty mudstone layer. Moreover, the contact boundary illustrated by the white dashed line between these two lithological units is delineated, being subparallel to the slope's surface without noticeable fluctuations.

As previously mentioned, GPR profile C ran from southeast to southwest along the stepwise flat arc trace and had an effective penetration depth of 7.5 m during the GPR surveys. Comparing the radar reflector signatures and their corresponding geological interpretations described in Figure 9a,b, we can see two central lithological units and a rugged bedrock structure in GPR profile C (Figure 9c). Similarly, the overlying material expressed by moderately continuous and darker reflectors relates to the colluvial layer composed of gravelly silty clay. In comparison, the underlying medium with brighter and discontinuous reflectors correlates with the bedrock of silty mudstone. From 0 m to 48 m in the horizontal distance, it seems that the boundary between the colluvium and bedrock fluctuates with notable troughs and crests, in agreement with the interpretation of ERT profile C (Figure 7a). The maximum and minimum depths to the bedrock occur at 16 m and 27 m, close to the 18 m and 28 m exhibited in ERT profile C in Figure 7a.

Conclusively, the GPR images better indicate the thickness of the topsoil and boundary from the underlying bedrock and show detailed information on the subsurface material, especially in the near-surface zone. However, they fail to detect the interfaces separating strongly weathered and rarely weathered rock, due to the strong attenuation of the radar waves penetrating fractured rocks.

### 3.3. Comparison between ERT and GPR Results

To further compare the ERT and GPR measurement results and hereby discuss the effectiveness and accuracy of these two combined geophysical methods in the delineation of subsurface characteristics, the inverted resistivity image and processed radargram of profile B were plotted and are shown in Figure 10. First, we have to point out that the errors in the topographical and lithological interpretations between the ERT and GPR images are unavoidable due to the differences in the manipulation and manifestation of databases using two geophysical technologies. The contrast both in the resistivity and the reflected pattern make it easier to see two stratigraphic layers with diverse features as described earlier in previous parts, showing an upper layer with a decreased trend in thickness from the crest to the toe of the slope, which is separated by a boundary being subparallel to the ground from the underlying bedrock. Obviously, in clay-rich conditions, GPR is able to distinguish the clay layer from mudstone layer by reflected profile when superimpos-

ing elevation values. This may be mainly because of the difference in internal structure (e.g., extent of consolidation) and compositions between clay and mudstone.

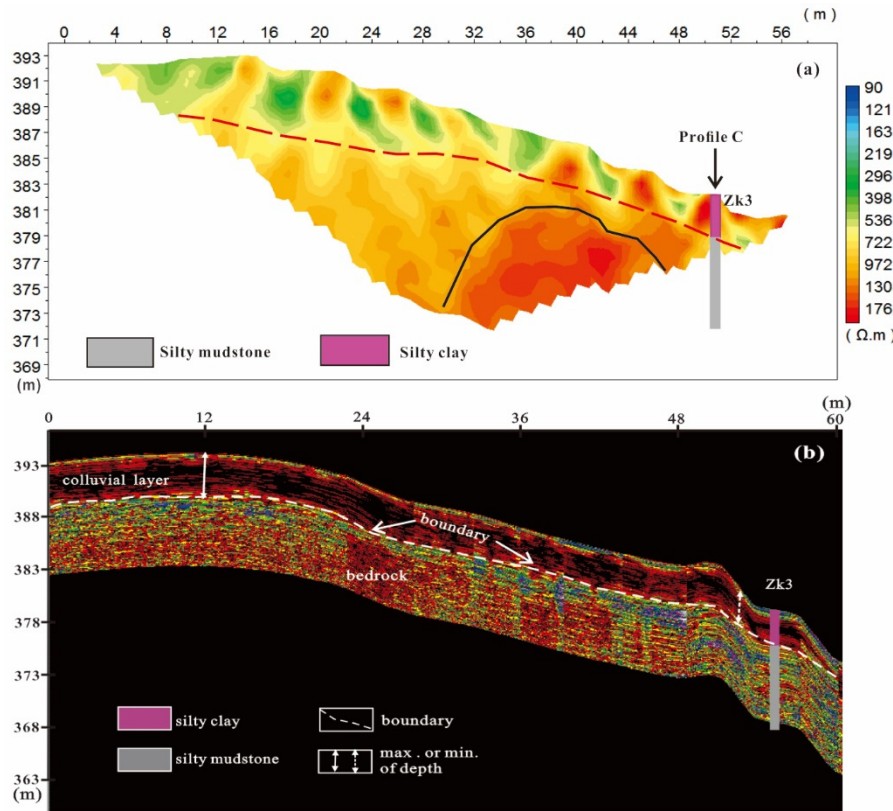

**Figure 10.** Comparison between ERT and GPR images of profile B. (**a**) ERT image of profile B, (**b**) GPR image of profile B.

Moreover, ERT is more likely to detect the area of rarely weathered bedrock between a distance of 30 and 48 m below the elevation of 381 m, whereas GPR fails to exhibit this phenomenon owing to its particular propagating ability. Commonly, the deeper the penetration depth, the poorer the resolution. In short, GPR is more accurate in reflecting shallow unknown objects (such as the planting soil layer in Figure 9a,b) and faster in gathering information, but it has a smaller probing depth than ERT.

### 3.4. Model of Potentially Unstable Body

Figure 11 shows the 3D interpreted model of a potentially unstable body built from the above analysis of the subsurface features through the combined ERT and GPR surveys as well as the borehole datasets. Obviously, 3D visualization of it enables us to extract essential information concerning the geometric shape, boundary, initial thickness, and lithological variation of the unstable body, which is important for its volume estimation. Indeed, the volume of the slide mass is one of the most important parameters for understanding its kinetic behavior and the potential effect on objects at risk. Suppose the surface area and average thickness are known. In that case, the volume of the potential landslide and its potential influence area can be rapidly calculated, the results of which are the fundamental basis for risk assessment and mitigation, and disaster protection and prevention.

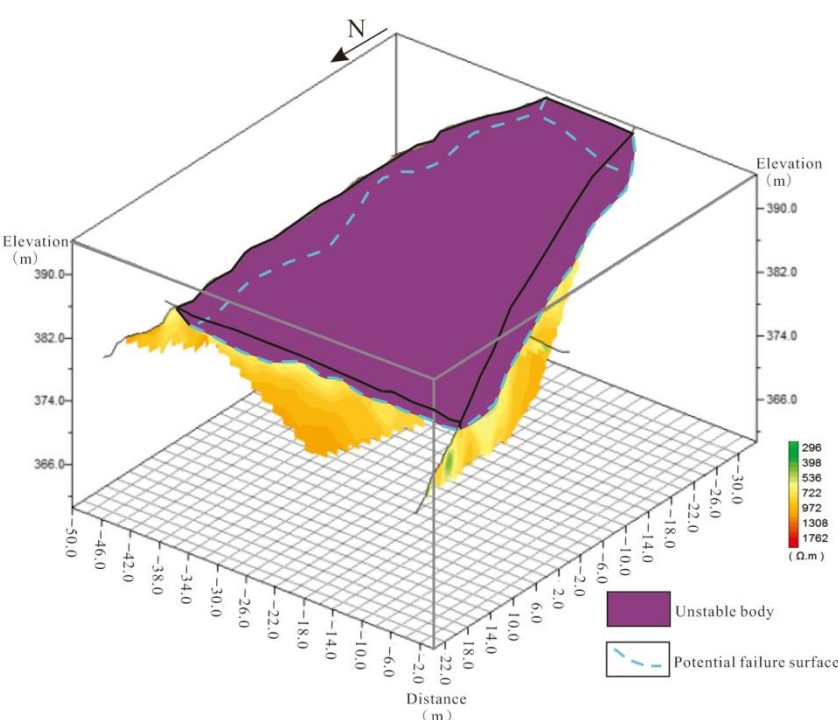

**Figure 11.** Three-dimensionally (3D) interpreted model of a potentially unstable body based on ERT and GPR images. The light blue dashed lines represent the boundary of the potential failure surface.

Further, the unstable mass's thickness is inhomogeneous and significantly undulates on the east part of the slope, as shown in Figure 11. Under this context, the commonly used drilling techniques are less likely to distinguish all these irregular distributions due to the insufficient survey points at the slope scale. The integrated preliminary and non-invasive geophysical surveys are particularly critical, as they can provide helpful references for the location selection of boreholes during geological investigations. It can also be inferred that the potential rupture surface might be located at the transition zone between the silty clay and the mudstone bedrock, probably in the heavily weathered rock (rich in clay, underground water, and cracks). Based on the identification of the potentially unstable body, more attention should be paid to the surface drainage to reduce the influence of rainfall infiltration in the typhoon season.

Although numerous studies have highlighted the inherent ambiguity of any individual geophysical method when interpreting unknown underground objects, they have also emphasized reducing uncertainty by incorporating multiple pieces of information acquired from diverse data sources, e.g., geophysical, geotechnical, and geological data. In the successful work published by [41], information on a colluvial layer, bedrock interface, potential sliding surface, and underground seepage system was illustrated by employing multi-geophysical methods and drilling data. Similarly, ref. [36] investigated the undulating topography of the bedrock beneath clay by integrating GPR and ERT data as well as boreholes. In this case study, the joint application of ERT and GPR was further confirmed to be capable of decreasing such uncertainty and applicable when delineating the subsurface features in clayey environments. Consequently, for Zhejiang Province where landslides mainly occur along the contact zone between unconsolidated deposits and the bedrock and are commonly shallow, the combination of ERT and GPR could be a reliable tool to obtain information about potential sources of risk. However, it is difficult to extract a precise threshold representing the potential failure zone in terms of the resistivity, amplitude, or energy from the above results.

## 4. Conclusions

Amid a changing climate, China is facing the specter of even more significant disaster risks in the future, which may also bring global cascading impacts. In Zhejiang Province, a large number of small hilly slopes are in an unstable state and have not been effectively investigated due to the difficulties of human surveys, the drawbacks of conventional geotechnical tools, and the limits of financial resources. This paper demonstrated how a combined geophysical method consisting of ERT and GPR with borehole data could effectively identify the subsurface characteristics of a clayey slope in a relatively short time, and how this delineated the potential unsteady body.

The inverted resistivity sections, radargram images, and single-channel waveform in terms of the amplitude/energy versus the depth indicated two lithological layers, consistent with the field observation and borehole data. The potential failure surface will most likely develop in the strongly weathered mudstone in the depth range of 3–7 m, and the average depth is 5 m. In addition, the thickness of the unstable mass is non-uniform in this slope, being much greater on its east and crest parts.

The GPR survey was suitable for identifying the shallow subsurface features with high-resolution imaging capability. Its reflection profile and the waveform of a single-channel GPR signal provided a valuable contribution to the analysis of the stratum structure and unstable body, even in a clay-rich environment. However, GPR could not distinguish the strongly weathered layer from the intermediary or rarely weathered layer due to their similarity in dielectric properties. The GPR wave was strongly attenuated when encountering anomalous stones or boulders in the topsoil layer. ERT could simultaneously assure both the resolution and exploration depth up to a maximum of 20 m and could be applied to detect the degree of weathering of the bedrock. The fresh bedrock was illustrated to exist at deeper zones in the ERT profiles. However, it was not easy to propose precise resistivity thresholds for different lithological layers.

In conclusion, the combined method of ERT and GPR was beneficial for fast field investigation of the subsurface features in a clay-rich condition, which could support a reference for geohazard prediction and prevention when precise knowledge of the subsurface is absent. It could also offer guidance for selecting borehole and trench locations.

**Author Contributions:** Conceptualization, Y.Y. (Yajing Yan) and Y.Y. (Yongshuai Yan); methodology, G.Z.; software, Y.Y. (Yongshuai Yan); validation, Y.Y. (Yongshuai Yan) and G.Z.; formal analysis, Y.Y. (Yajing Yan); writing—original draft preparation, Y.Y. (Yajing Yan); writing—review and editing, Y.Z.; supervision, Z.W. All authors have read and agreed to the published version of the manuscript.

**Funding:** This research was funded by the Open Research Fund of Key Laboratory of Subsurface of Hydrology and Ecological Effect in Arid Region (Chang'an University), Ministry of Education (No. 300102290503), Ministry of Education, and the Belt and Road Special Foundation of the State Key Laboratory of Hydrology—Water Resources and Hydraulic Engineering (No. 2021490511).

**Institutional Review Board Statement:** Not applicable.

**Informed Consent Statement:** Not applicable.

**Data Availability Statement:** Not applicable.

**Acknowledgments:** The authors would like to express their heartfelt gratitude to the anonymous reviewers for their constructive comments that were valuable for improving this manuscript.

**Conflicts of Interest:** The authors declare no conflict of interest.

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
