# Peer review of "Combined ERT and GPR Data for Subsurface Characterization of Weathered Hilly Slope: A Case Study in Zhejiang Province, Southeast China"

_sustainability, doi:10.3390/su14137616_

Round 1

Reviewer 1 Report

The manuscript is a case study in Zhejiang, China, to investigate the surface characteristics of clayey slope using ERT and GPR. This work is complete, well organised and clearly written. My only concern is the innovation, as the joint application of the two methods is not the first time it has been proposed, are there any new findings, or is this just a repeat?

How to understand "integrated" in the title? I think the work seems only joint application rather than the integration of the two methods.

Author Response

Thanks for the constructive suggestions and comments, details are supported in the Word profile.

Reviewer 2 Report

Journal
Sustainability (ISSN 2071-1050)
Manuscript ID
sustainability-1738279
Type
Article
Title
Integrated ERT and GPR Data for Subsurface Characterization of Clayey Slope: A Case Study in Zhejiang Province, Southeast China

Review Comments
This paper presents a case study in Zhejiang Province, Southeast China. The study is about subsurface characterization of clayey Slope and rainfall induced landslides. Study explains the importance of geophysical studies on natural slopes in Zhejiang, China and rainfall induced landslide vulnerable areas as well. Overall, the study is interesting. However, there are still major issues that should be resolved before the manuscript can be accepted.

1. Authors conducted geophysical tests (Ground Penetrating Radar - GPR and Electrical Resistivity Tomography - ERT), is there any code of practice or any International/National standards followed during testing? If so, mention that standards in manuscript.
2. Limitations of GPR test in slope/landslide study should be mentioned.
3. Authors should explain what are the disadvantages of conventional geotechnical techniques? and mention in manuscript. (Line 83)
4. Is the study area (Yuxi village in Taishun county, Wenzhou) witnessed any landslides in the past? If so, mention about the history and type and details of landslides particularly for that site.
5. Authors mentioned the slope having clayey soil at top portion (top layer). Kindly provide index and engineering properties of clayey soil.
6. Authors conducted bore hole investigation at three points (ZK1, ZK2, ZK3), but not shown the basic details like thickness of soil and rock layers, provide the bore log for the three locations (Line 166 – 168)
7. The authors must explain the novelty and importance of the study to the international research and academic community.
8. Mention mineralogy and weathering characteristics of weathered mudstone.
9. Justification of selection of the site (like vulnerability, potential infrastructural damages, existence of life line structures etc.) needs to be provided.
10. Tests on rock has to be conducted and results to be presented. Then only, comparison with geophysical tests will be more meaningful.

Author Response

Thanks for the constructive suggestions and comments, details are shown in the Word profile.

Reviewer 3 Report

In the manuscript, the authors presented a case study of subsurface characterization of clayey slope using integrated geophysical methods.  However, to be published in the journal, it would be preferable to add some  minor revisions and improvements according to the comments listed below;

1.The authors should change the title to 'Integrated ERT and GPR Data for Subsurface Characterization of Weathered Hilly Slope: A Case Study in Zhejiang Province, Southeast China'.

2.Though the authors used clayey slope in title, keywords, the physical representation and geophysical characterization does not represent the same. The authors need to provide better clarification on this.

3. The resistivity results should be compared with those of the existing literature, to make the characterization more credible.

4. The citation in the main text need to be correctly formatted. For example; Page 13; Line 380; Precisely, the resistivity value for silts is about 10-2000 Ohm.m and 1-100 Ohm.m for 379 clay as well as 100-1400 Ohm.m for gravel, while >1000 Ohm.m for rocks (M.C. Diallo et 380 al., 2019; Ahmad Afshar et al., 2015).

5. Authors need to verify through out the manuscript for grammatical and typical errors. For example, Page 9; Line 246; it is roughly perpendicular to the slope's other two profiles and dip direction of. Some interesting conclusions were illustrated, but a minor revision is necessary before it can be accepted.

Author Response

(The authors gave the same response as above.)

Round 2

Reviewer 2 Report

Authors have incorporated the answers to all the comments.